# Exploring health anxiety and its association with quality of life in Yemeni medical undergraduates: A national cross-sectional

Mohamed Terra[1☉*], Mohamed Baklola[2☉*], Naji Al-bawah[3☉*],
Husam Addin Ban Rafuid[4], Ziad Mohammed AL-Othrubi[3], Hadeel Z. Mohammed[5],
Ehab Sharyan[3], Mohammed R. Arrabyee[3], Mohand Almarebi[6], Akram Arrabyee[7],
Fahd Alsameeai[8], Sadiq Altbali[9], Amira Yasmine Benmelouka[10]

**1** General Practitioner, Ministry of Health, Cairo, Egypt, **2** Intern Doctor, Mansoura University Hospitals, Mansoura, Egypt, **3** Faculty of Medicine, Sana'a University, Sana'a, Yemen, **4** Faculty of Medicine, Emirates International University, Sana'a, Yemen, **5** Faculty of Medicine, Aden University, Aden, Yemen, **6** Faculty of Medicine, Dhamar University, Dhamar, Yemen, **7** Faculty of Pharmacy, Alnasser University, Sana'a, Yemen, **8** Faculty of Medicine, 21 University, Sana'a, Yemen, **9** Faculty of Medicine, Ibb University, Ibb, Yemen, **10** Faculty of Medicine, University of Algiers, Algiers, Algeria

☉ These authors contributed equally to this work.
* Mohamedtera75@gmail.com (MT); Mohamedbaklola2000@gmail.com (MB);
Najialbawah@gmail.com (NA)

## Abstract

### Background

Health anxiety is an increasingly recognized concern among medical students, who are frequently exposed to health-related information and academic stressors. This study aimed to estimate the prevalence of health anxiety among Yemeni medical undergraduates and to examine its association with quality of life (QoL) and selected demographic and academic factors.

### Methods

A national cross-sectional study was conducted between February and April 2025 across ten medical schools in Yemen. A total of 2,573 undergraduate medical students completed an anonymous online questionnaire assessing sociodemographic and academic characteristics, health anxiety using the Short Health Anxiety Inventory (SHAI), self-perceived health, and quality of life using the SF-12 survey. Data were analyzed using descriptive statistics, Pearson's Chi-square test, the Mann–Whitney U test, Pearson's correlation, and multivariable logistic regression.

### Results

Clinically significant health anxiety (SHAI ≥27) was observed in 9.8% of participants. Health anxiety was more prevalent among female students and those residing in rural areas (p = 0.010 and p < 0.001, respectively). Lower academic performance,

**Data availability statement:** The minimal underlying dataset supporting the findings of this study is publicly available in the Figshare repository at https://doi.org/10.6084/m9.figshare.30217726.v1.

**Funding:** The author(s) received no specific funding for this work.

**Competing interests:** The authors have declared that no competing interests exist.

**Abbreviations:** HA, Health anxiety; QoL; Quality of life; SHAI, Short Health anxiety Inventory.

dissatisfaction with academic progress, and personal or family history of psychiatric disorders were significantly associated with health anxiety. Students with clinically significant health anxiety reported lower quality of life scores (median = 27; IQR: 24–29) compared with those without health anxiety (p < 0.001). In multivariable analysis, male gender, urban residence, and academic satisfaction were independently associated with lower odds of health anxiety, whereas low academic performance and psychiatric history were associated with higher odds. A moderate negative correlation was observed between health anxiety and quality of life (r = −0.37, p < 0.001).

## Conclusion

Nearly one in ten Yemeni medical students exhibited clinically significant health anxiety, which was associated with poorer quality of life and unfavorable academic and psychological characteristics. These findings highlight the importance of integrating mental health support and academic counseling services within medical schools to promote student well-being.

## Introduction

Anxiety disorders are among the most prevalent psychiatric conditions globally, affecting approximately 7.3% of the population [1]. Within this broad category, health anxiety, which is characterized by excessive worry about having or developing a serious illness, has emerged as a prevalent issue among medical students [2]. Although often used interchangeably with hypochondriasis, health anxiety encompasses a spectrum of cognitive and behavioral patterns, including heightened focus on bodily sensations and misinterpretation of these as signs of illness, even in the absence of clinical symptoms [3,4].

Medical students are especially vulnerable to health anxiety due to their continuous exposure to medical information and diseases throughout their training. The process of learning about complex pathologies can lead to misattribution of normal bodily sensations to serious health conditions, a phenomenon sometimes referred to as "medical student syndrome" [5]. This persistent preoccupation can cause substantial psychological distress, impair daily functioning, and contribute to depressive symptoms and even delusional thinking in severe cases [6].

Individuals experiencing health anxiety may adopt maladaptive coping strategies, such as frequent medical consultations, unnecessary diagnostic testing, or compulsive online health-related searches—often referred to as "cyberchondria" [2,7,8]. These behaviors can further reinforce anxiety and disrupt social relationships, academic engagement, and overall well-being.

For medical students, health anxiety poses unique challenges that significantly compromise their quality of life across multiple domains. The intensive nature of medical education, with continuous exposure to disease information through lectures and clinical rotations, creates an environment where health anxiety can

be particularly amplified, often manifesting as "medical student syndrome" [9]. This persistent preoccupation with personal health concerns consumes mental energy critically needed for the demanding cognitive load of medical training, interfering with concentration during lectures, reducing study efficiency, and creating distress during clinical encounters [10].

Cyberchondria behaviors become especially problematic in this population, as medical students possess enough knowledge to understand medical terminology but may lack clinical experience to properly contextualize symptoms, leading to anxiety-driven internet searches that can spiral into hours of unproductive worry [10].

Several international studies have highlighted the high prevalence of health anxiety among medical students. In India, approximately one in seven medical students exhibited symptoms of health anxiety, with preclinical students particularly affected [9]. Similarly, a study conducted in the United Arab Emirates reported that nearly 10% of undergraduate students experienced health anxiety, often associated with prior physical or psychiatric illnesses that was linked to their future career outlook [11]. In Egypt, a multi-center study conducted across ten universities found that 15.7% of participants experienced clinically significant health anxiety. The prevalence was notably higher among female students (17.5%) and those who were dissatisfied with their academic performance (18%) [12]. Additionally, research at Tanta University in Egypt reported a high prevalence of hypochondriasis, particularly among female and fourth-year medical students [13]. Further studies have emphasized that medical students often exhibit elevated levels of moral distress and mental health literacy compared to their non-medical peers, likely due to their enhanced awareness and understanding of disease processes [14].

In Yemen, anxiety disorders appear to be particularly prevalent among medical students. A study conducted in 2024 reported that 34.8% of Yemeni medical students experienced symptoms of anxiety [15]. However, despite this high prevalence, there remains a notable gap in the literature regarding health anxiety specifically. To date, no published studies have investigated the prevalence or implications of health anxiety among medical students in Yemen. Given the academic and psychological challenges faced by this population, it is essential to examine how health anxiety may affect their well-being. Therefore, this study aims to assess the prevalence of clinically significant health anxiety among Yemeni medical students and explore its association with their quality of life.

## Methods

### Study setting and duration

A national, multi-institutional study was conducted across ten medical schools in Yemen over a three-month period, from February 10 to April 10, 2025, following ethical approval. The participating institutions were selected to ensure representation of different geographic regions and educational settings within the country.

### Study design

This study employed a descriptive cross-sectional design with an analytical component. This design enabled the estimation of the prevalence of health anxiety among medical students and the examination of its associations with quality of life and selected demographic and academic variables. Due to the cross-sectional nature of the study, causal relationships could not be inferred.

### Study population

The study population comprised undergraduate medical students enrolled in Yemeni medical schools. Students from all academic years (first through sixth year) were eligible to participate. Inclusion criteria were current enrollment in a medical program and voluntary agreement to participate. No exclusion criteria were applied beyond incomplete questionnaire responses.

## Sample size calculation

The sample size calculation was informed by a previously reported prevalence of health anxiety among medical students, estimated at 15.7% [12]. Assuming a 95% confidence level, a 5% margin of error, and a statistical power of 80%, the minimum required sample size was estimated. To account for institutional variability across multiple universities, a design effect of 1.0 was applied, resulting in a minimum required sample size of 2,020 participants. Ultimately, 2,573 students completed the survey, exceeding the calculated requirement and thereby enhancing the precision of the estimates.

## Study instruments

Data were collected using a structured, self-administered online questionnaire developed specifically for this study. The questionnaire consisted of four sections:

### Sociodemographic and academic characteristics

The first section collected information on age, gender, place of residence (urban or rural), academic year, most recent semester academic performance, satisfaction with academic performance, presence of family members in the medical field, and personal or family history of psychiatric disorders. Participants were also asked whether they experienced external pressure to pursue medical education.

### Health anxiety assessment

Health anxiety was assessed using the 18-item Short Health Anxiety Inventory (SHAI) [16], a widely used instrument designed to measure anxiety related to health concerns independent of physical illness. The SHAI comprises two subscales assessing general health anxiety and perceived negative consequences of illness. Items are scored on a four-point Likert scale (0–3), yielding a total score ranging from 0 to 54. A cutoff score of ≥27 was used to indicate clinically significant health anxiety, in line with prior validation studies [16]. The Arabic version of the SHAI has demonstrated good internal consistency (Cronbach's $\alpha = 0.77$–$0.85$ for subscales; $\alpha = 0.85$ overall) and confirmed construct validity, supporting its suitability for use in this population.

### Self-perceived health status

The third section included an open-ended question asking participants to report any illnesses or medical conditions they believed they might have. This item was included to capture subjective health perceptions, which may influence health-related anxiety.

### Quality of life assessment

Quality of life was measured using the Arabic version of the 12-Item Short Form Health Survey (SF-12, version 2) [17]. The SF-12 evaluates health-related quality of life across eight domains encompassing physical and mental health components. Higher scores indicate better perceived quality of life. The Arabic SF-12 has demonstrated strong psychometric properties, including good internal consistency (Cronbach's $\alpha = 0.70$–$0.84$ for domains; $\alpha = 0.80$ overall), excellent test–retest reliability (intraclass correlation coefficient $= 0.80$–$0.99$), and validated construct structure.

## Sampling strategy

A non-probability convenience sampling approach was used to recruit participants from the ten participating medical schools. Probability-based sampling was not feasible due to logistical constraints, including the absence of centralized student enrollment lists, wide geographic dispersion of institutions, limited resources, and security challenges within the country. To enhance sample heterogeneity, students from different academic years, genders, and geographic

 

backgrounds were invited to participate. Although convenience sampling limits representativeness, the large sample size and multi-institutional recruitment partially mitigate this limitation.

### Data collection procedure

Data were collected electronically using a Google Forms-based questionnaire. The survey link was disseminated through multiple official and widely used student communication channels, including university-affiliated Telegram groups and social media platforms, to reduce reliance on a single recruitment source. Participation was voluntary and anonymous, and no incentives were provided. Informed consent was obtained electronically prior to survey submission. The anonymous and self-administered format was intended to encourage honest reporting and reduce social desirability bias.

### Statistical analysis

Data were analyzed using IBM SPSS Statistics version 27. Descriptive statistics were used to summarize participant characteristics. Continuous variables were reported as means with standard deviations or medians with interquartile ranges, depending on data distribution, while categorical variables were presented as frequencies and percentages. Associations between categorical variables and the presence of clinically significant health anxiety were examined using Pearson's Chi-square test. For continuous variables that did not meet normality assumptions, group comparisons were conducted using the Mann–Whitney U test or Wilcoxon rank-sum test, as appropriate.

The relationship between health anxiety and quality of life was evaluated using Pearson's correlation coefficient. Binary logistic regression analysis was performed to identify independent predictors of clinically significant health anxiety. Variables with a p-value <0.05 in univariate analyses were entered into the multivariable model. Adjusted odds ratios (AORs) with 95% confidence intervals (CIs) were reported. Statistical significance was set at $p \leq 0.05$ for all analyses.

### Ethical approval and consent to participate

The study procedures were conducted by the ethical standards outlined in the Declaration of Helsinki and its later revisions. Ethical clearance was granted by the Institutional Review Board of Sana'a University (Approval No.231). Participation was completely voluntary and anonymous, with no incentives provided. A detailed explanation of the study's purpose was presented on the initial page of the online questionnaire. Informed consent was obtained electronically from all participants before their involvement, and they were informed of their right to withdraw at any time without any repercussions.

## Results

### Prevalence of health anxiety

Among the 2,573 Yemeni medical students included in the analysis, 251 participants (9.8%) met the criteria for clinically significant health anxiety based on the Short Health Anxiety Inventory (SHAI) cutoff score of ≥27.

### Association between demographic characteristics and Health anxiety

Several demographic and academic variables were significantly associated with health anxiety (Table 1). Gender showed a significant association, with a higher prevalence observed among female students (11.5%) compared with male students (8.5%) (p = 0.010). Residence was also significantly associated with health anxiety, with a higher prevalence among students residing in rural areas (14.9%) compared with those in urban areas (8.6%) (p < 0.001).

Academic level was not significantly associated with health anxiety (p = 0.30). In contrast, academic performance variables demonstrated significant associations. Students with a last semester score below 75% exhibited a higher prevalence of health anxiety (15.2%) compared with those scoring 75–90% (9.2%) or above 90% (8.9%) (p = 0.003). Similarly,

**Table 1. Demographic characteristics and prevalence of health anxiety among students.**

| Characteristic | | No health anxiety (n = 2,322) | Health anxiety (n = 251) | P-value |
|---|---|---|---|---|
| **Age** | | 22 (20-24) | 22 (20-24) | 0.40 |
| **Academic level** | Year 1 | 341 (92%) | 29 (7.8%) | 0.30 |
| | Year 2 | 472 (90%) | 54 (10%) | |
| | Year 3 | 484 (90%) | 52 (9.7%) | |
| | Year 4 | 390 (88%) | 54 (12%) | |
| | Year 5 | 366 (90%) | 40 (9.9%) | |
| | Year 6 | 269 (92%) | 22 (7.6%) | |
| **Gender** | Female | 968 (88%) | 126 (12%) | **0.01** |
| | Male | 1,354 (92%) | 125 (8.5%) | |
| **Residence** | Rural (Countryside) | 406 (17%) | 71 (28%) | **< 0.001** |
| | Urban (City) | 1,916 (83%) | 180 (72%) | |
| **University type** | Governmental University | 406 (85%) | 71 (15% | 0.11 |
| | Private University | 1,916 (91%) | 180 (8.6%) | |
| **Last semester score (Percentage)** | Less than 75% | 251 (85%) | 45 (15%) | **0.03** |
| | 75% − 90% | 1,299 (91%) | 131 (9.2%) | |
| | More than 90% | 772 (91%) | 75 (8.9%) | |
| **Satisfaction with academic performance** | No, I am not satisfied. | 892 (88%) | 121 (12%) | **0.03** |
| | Yes, I am satisfied. | 1,430 (92%) | 130 (8.3%) | |
| **Having family members in medical field** | No, I don't. | 1,103 (90%) | 116 (9.5% | 0.70 |
| | Yes, I do. | 1,219 (90%) | 135 (10.0%) | |
| **Family history of psychiatric disorders** | No, I don't. | 1,466 (92%) | 127 (8.0%) | **< 0.001** |
| | Yes, I do. | 856 (87%) | 124 (13%) | |
| **Past history of psychiatric disorders** | No, I don't. | 888 (94.5%) | 52 (5.5%) | **< 0.001** |
| | Yes, I do. | 1,434 (88%) | 199 (12%) | |

Data are presented as median (Q1–Q3) for continuous variables and n (%) for categorical variables. Percentages represent column proportions. P-values were calculated using the Wilcoxon rank-sum test for continuous variables and Pearson's Chi-square test for categorical variables. Statistical significance was set at p ≤ 0.05.

dissatisfaction with academic performance was associated with a higher prevalence of health anxiety (12.0%) compared with satisfaction (8.3%) (p = 0.003).

Having family members in the medical field was not significantly associated with health anxiety (p = 0.70). However, both family history and personal history of psychiatric disorders were significantly associated with health anxiety. Students with a family history of psychiatric disorders had a higher prevalence of health anxiety (12.7%) compared with those without such a history (8.0%) (p < 0.001). Likewise, students with a personal history of psychiatric disorders showed a higher prevalence (12.2%) compared with those without (5.5%) (p < 0.001).

### Quality of life and participant characteristics

Quality of life, assessed using the SF-12 total score, varied significantly across several demographic and academic variables (Table 2). Academic level was significantly associated with quality of life (p < 0.001), with sixth-year students reporting the highest median score (30; IQR: 26–35). Male students had higher median quality of life scores (29; IQR: 27–33) compared with female students (28; IQR: 26–31) (p < 0.001).

**Table 2. Quality of life (SF-12 total score) across sociodemographic and academic characteristics.**

| Characteristic | | Quality of Life | | | P-value |
|---|---|---|---|---|---|
| | | Median | Q1 | Q3 | |
| Academic Level | Year 1 | 29 | 26 | 32 | **< 0.001** |
| | Year 2 | 28 | 26 | 31 | |
| | Year 3 | 29 | 27 | 33 | |
| | Year 4 | 29 | 26 | 32 | |
| | Year 5 | 29 | 26 | 33 | |
| | Year 6 | 30 | 26 | 35 | |
| Gender | Female | 28 | 26 | 31 | **< 0.001** |
| | Male | 29 | 27 | 33 | |
| Residence | Rural (Countryside) | 29 | 26 | 32 | 0.30 |
| | Urban (City) | 29 | 26 | 32 | |
| University Type | Government University | 29 | 26 | 33 | **0.04** |
| | Private University | 29 | 26 | 32 | |
| Last Semester Score (Percentage) | Less than 75% | 28 | 25 | 31 | **< 0.001** |
| | 75% – 90% | 29 | 26 | 33 | |
| | More than 90% | 28 | 25 | 31 | |
| Satisfaction with Academic Performance | No, I am not satisfied. | 28 | 26 | 32 | **< 0.001** |
| | Yes, I am satisfied. | 29 | 26 | 33 | |
| Having Family Members in the Medical Field | No, I don't. | 29 | 26 | 33 | 0.30 |
| | Yes, I do. | 29 | 26 | 32 | |
| Family History of Psychiatric Disorders | No, I don't. | 29 | 26 | 33 | **< 0.001** |
| | Yes, I do. | 28 | 26 | 31 | |
| History of psychiatric Disorders | | 28 | 26 | 31 | **< 0.001** |
| Health anxiety | | 27 | 24 | 29 | **< 0.001** |

Quality of life was assessed using the SF-12 total score; higher scores indicate better quality of life. Data are presented as median and interquartile range (Q1–Q3). P-values were calculated using the Wilcoxon rank-sum test. Statistical significance was defined as $p \leq 0.05$.

Residence was not significantly associated with quality of life (p = 0.30). However, students enrolled in governmental universities reported slightly higher quality of life scores than those in private universities (p = 0.04). Academic performance was significantly associated with quality of life (p < 0.001), with students scoring 75–90% in the last semester reporting the highest median scores. Students satisfied with their academic performance also demonstrated higher quality of life compared with those who were dissatisfied (p < 0.001).

No significant differences in quality of life were observed based on having family members in the medical field (p = 0.30). In contrast, students with a family history or a personal history of psychiatric disorders reported significantly lower quality of life scores (both p < 0.001). Students with clinically significant health anxiety had markedly lower quality of life (median = 27; IQR: 24–29) compared with those without health anxiety (p < 0.001).

## Predictors of health anxiety

Multivariable logistic regression analysis identified several factors independently associated with clinically significant health anxiety (Table 3). Male gender (AOR = 0.75; p = 0.038) and urban residence (AOR = 0.54; p < 0.001) were associated with lower odds of health anxiety. Conversely, a last semester score below 75% was associated with higher odds of health anxiety (AOR = 1.50; p = 0.036), while scores above 90% were not significantly associated (p = 0.938).

**Table 3. Multivariable logistic regression analysis of factors associated with clinically significant health anxiety.**

| Variable | Odds Ratio (AOR) | 95% CI | P-value |
|---|---|---|---|
| Gender (**Male**) | 0.752 | [0.575 - 0.984] | **0.038** |
| Residence (**Urban**) | 0.535 | [0.397 - 0.727] | **<0.001** |
| Last Semester Percentage (**Less than**...) | 1.5 | [1.02 - 2.16] | **0.036** |
| Last Semester Percentage (**More than**...) | 1.01 | [0.745 - 1.37] | 0.938 |
| Satisfied Academic Performance (**Yes**) | 0.759 | [0.578 - 0.998] | **0.048** |
| Family History Psychiatric Disorder (**Yes**) | 1.55 | [1.19 - 2.02] | **0.001** |
| Past Psychiatric History (**Yes**) | 2.1 | [1.54 - 2.93] | **<0.001** |

Clinically significant health anxiety was defined as a Short Health Anxiety Inventory (SHAI) score ≥27. Reference categories were female gender, rural residence, last semester score 75–90%, dissatisfaction with academic performance, and absence of personal or family psychiatric history. Variables included in the model were those significant in univariate analyses. Adjusted odds ratios (AORs) with 95% confidence intervals (CIs) are reported. Statistical significance was set at p ≤ 0.05.

Dissatisfaction with academic performance was associated with higher odds of health anxiety (AOR = 0.76; p = 0.048). Additionally, both family history of psychiatric disorders (AOR = 1.55; p = 0.001) and personal history of psychiatric disorders (AOR = 2.10; p < 0.001) were independently associated with health anxiety.

## Correlation between health anxiety and quality of life

A statistically significant negative correlation was observed between health anxiety and quality of life (r = −0.37, p < 0.001), indicating that higher health anxiety scores were associated with lower quality of life among participants (Fig 1).

## Discussion

This national study is the first to examine health anxiety and its relationship with quality of life (QoL) among Yemeni medical students. Using the Short Health Anxiety Inventory (SHAI), we found that 9.8% of participants met criteria for clinically significant HA. This prevalence aligns closely with studies from the United Arab Emirates (9.3%) and Pakistan (11.9%) [18,19], and is lower than that reported in Egypt (15.7%) and Saudi Arabia (30%) [12,20,21]. These variations highlight the contextual influences shaping students' psychological well-being, including cultural norms, access to mental health resources, academic pressures, and the stability of the educational environment. In Yemen, the prolonged humanitarian crisis, limited psychological services, and unstable educational conditions may contribute to elevated stress; however, the moderate prevalence observed should be interpreted cautiously and may reflect underreporting related to stigma surrounding mental health discussions [22,23].

Consistent with prior research in Egypt, Pakistan, and Saudi Arabia [12,19–21], our findings revealed a higher prevalence of HA among female students. This association may reflect the interaction of sociocultural, psychological, and academic factors rather than a single explanatory mechanism. Female students in the MENA region often face compounded stressors, including societal expectations, restricted autonomy, and heightened concern about academic and career prospects [24,25]. Additionally, women generally exhibit greater health awareness and are more likely to monitor physical sensations, predisposing them to health-related worries. Epidemiological evidence further supports that anxiety disorders are more prevalent among women globally [24,25]. Within the Yemeni context, cultural constraints and social expectations may further intensify these vulnerabilities, potentially contributing to the observed gender differences in HA [26].

Academic satisfaction and performance emerged as strong predictors of HA. Students who were dissatisfied with their academic progress or had lower grades were significantly more likely to experience HA. This relationship is consistent with findings from Egypt and Saudi Arabia [12,20,21], where dissatisfaction and poor academic achievement were major stressors. The medical curriculum's intensity, frequent exposure to illness-related material, and limited support

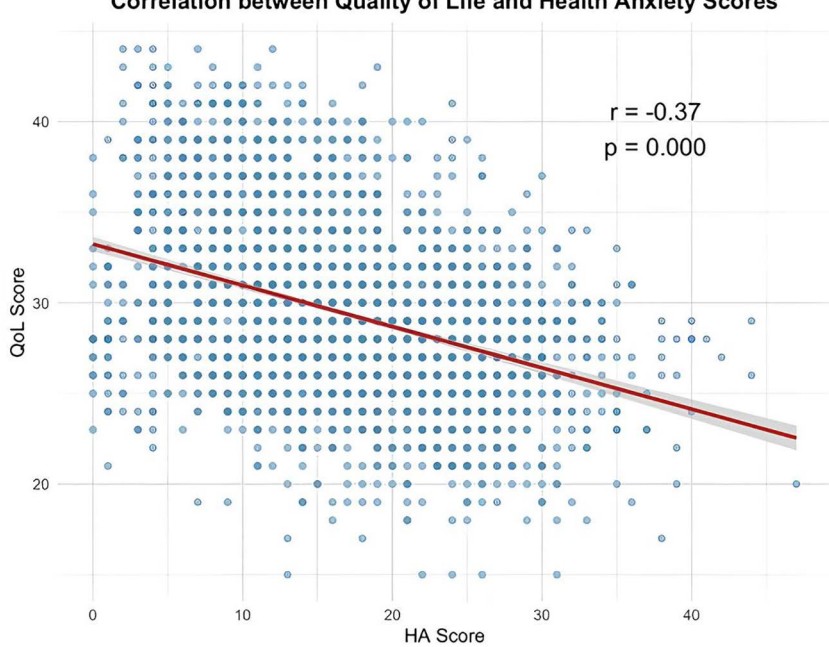

**Fig 1. Scatter plot of quality of life and health anxiety among participants, highlighting negative correlation.**

mechanisms can heighten psychological stress, which may be associated with maladaptive health-related concerns rather than directly causing them [27–30]. In the Yemeni context, disrupted educational continuity, irregular academic schedules, and scarce institutional support likely amplify these pressures [31]. Moreover, dissatisfaction with performance may not only reflect academic difficulties but also internalized perfectionism, a trait previously linked to increased health anxiety and reduced well-being among medical trainees [32,33].

Interestingly, our data did not reveal significant across academic years, mirroring the non-significant patterns reported in Pakistan and Egypt [12,19]. This suggests that health anxiety may persist throughout medical training rather than being confined to specific academic phases, potentially influenced by stable individual characteristics and broader contextual stressors.

Both personal and family histories of psychiatric disorders were strong predictors of HA, aligning with the broader literature [34–36]. Students with such histories may possess greater emotional sensitivity and maladaptive coping styles, making them more prone to interpret normal bodily sensations as pathological [37]. In settings with limited access to counseling or psychiatric care, such tendencies may remain unaddressed, reinforcing anxiety-related cognitive patterns. These findings highlight the importance of early identification and targeted psychological support for students with known psychiatric vulnerabilities [38].

A moderate negative correlation ($r = -0.37$) was observed between HA and QoL, indicating an inverse association rather than a causal relationship. This finding parallels those reported in Egypt and Saudi Arabia [12,20,21], where students with HA exhibited poorer QoL across both physical and psychological domains. The mechanism underlying this association may involve chronic worry and hypervigilance to bodily sensations, leading to sleep disturbances, impaired concentration, and reduced social and academic engagement [32,33,9]. Over time, such patterns may be associated with diminished psychological resilience and lower overall life satisfaction.

In Yemen, where medical students already face systemic instability, economic hardship, and limited institutional support, the coexistence of HA and reduced QoL is particularly concerning [39]. This underscores the urgent need for psychosocial interventions embedded within medical education frameworks.

When compared regionally, Yemen's HA prevalence appears moderate, lower than Egypt and Saudi Arabia but comparable to other MENA and South Asian countries. This variation may partly reflect differences in measurement tools and sampling approaches; however, contextual and cultural factors likely also play a role. Lower reported prevalence in Yemen may stem from under recognition or reluctance to disclose anxiety symptoms [40]. Conversely, in Saudi Arabia, greater academic intensity, financial pressure within private universities, and differences in institutional support systems have been proposed as contributing factors to higher HA rates [20,21].

Similarly, consistent patterns across countries, including female vulnerability, the role of academic dissatisfaction, and the inverse link with QoL, suggest that health anxiety among medical students is a transnational issue influenced by the interplay of personal predispositions, educational stress, and cultural context rather than by geography alone [41]. These converging findings support the relevance of our results beyond a single national context, while acknowledging limitations in generalizability.

The findings of this study have important practical implications. First, the integration of structured mental health support within Yemeni medical schools is warranted. Early screening, resilience-building interventions, and accessible counseling services may help mitigate health anxiety. Second, embedding mental health literacy and stress management within medical curricula could normalize help-seeking behaviors and reduce stigma. Third, academic mentoring and peer-support initiatives may alleviate performance-related stress, a modifiable factor consistently associated with HA in this and other studies. Given Yemen's socioeconomic constraints, low-cost and scalable strategies, such as peer-led support groups, online counseling platforms, and faculty training, may represent feasible and impactful interventions [42]. Policymakers and academic institutions should prioritize strengthening student welfare frameworks to support both academic success and long-term professional well-being.

## Limitations

Several limitations of this study should be acknowledged. First, the cross-sectional design restricts the ability to establish temporal or causal relationships between health anxiety and quality of life; therefore, the observed associations should be interpreted as correlational rather than causal. Second, data were collected using self-administered, online questionnaires, which may be subject to reporting and recall bias. Although anonymity, voluntary participation, and the use of validated psychometric instruments were intended to minimize social desirability bias, self-reported measures cannot fully substitute for clinical assessments.

Third, a non-probability convenience sampling strategy was employed, which may limit the external validity and generalizability of the findings to all Yemeni medical students. However, this approach was adopted due to practical and logistical constraints, including the absence of centralized student registries, wide geographic dispersion of institutions, and security challenges. To partially mitigate this limitation, a large sample was recruited from ten medical schools representing diverse geographic regions and academic settings. Fourth, the assessment of health anxiety relied on the Short Health Anxiety Inventory rather than structured clinical interviews, which may have resulted in misclassification. Nevertheless, the SHAI is a widely used, psychometrically validated instrument that has demonstrated reliability and validity in Arabic-speaking populations.

Fifth, although health anxiety and quality of life were assessed using validated tools, reliance on two primary quantitative instruments may not fully capture the complexity of the relationship between psychological distress and well-being. The inclusion of additional measures, such as perceived stress, coping strategies, academic workload, or qualitative approaches, could provide a more comprehensive understanding of these dynamics. Finally, unmeasured confounding variables, including broader psychosocial stressors related to the humanitarian and educational context in Yemen, may

have influenced participants' responses and were not examined in detail. Future longitudinal and mixed-methods studies incorporating clinical interviews and broader psychosocial assessments are warranted to build on these findings and enhance causal inference.

## Conclusion

This study revealed that approximately one in ten Yemeni medical students met criteria for clinically significant health anxiety, highlighting an important mental health concern within this population. Health anxiety was associated with several demographic and academic characteristics, including gender, residence, academic performance, and personal or family history of psychiatric disorders. Furthermore, a significant inverse association was observed between health anxiety and quality of life, indicating that students with higher levels of health anxiety reported poorer overall well-being.

Although causal relationships cannot be inferred due to the cross-sectional design, these findings underscore the relevance of health anxiety as a factor linked to diminished quality of life among medical students. Integrating targeted mental health support, early screening, and academic counseling within Yemeni medical schools may help address this burden. Such interventions could support students' psychological well-being, enhance academic engagement, and contribute to the development of a healthier learning environment, while promoting resilience among future healthcare professionals.

## Acknowledgments

We would like to express our deepest gratitude to the Yemeni medical student community for their invaluable support and active participation in this research. Their collaboration, commitment, and enthusiasm toward advancing medical knowledge were fundamental to the success of this study.

Our heartfelt thanks go to Professor Mohammed Alshehri and Abdul Wahab Al-Mutahar for their exceptional mentorship and unwavering support throughout this project. His insightful guidance, constructive feedback, and continuous encouragement greatly enriched the quality and direction of our work.

Special thanks are extended to the following individuals who contributed to the data collection phase in a supporting capacity:

Saif Alaribi and Yusra Rashed Al-Subaihi. Their assistance, though in minor roles, played a meaningful part in the completion of this study

## Author contributions

**Conceptualization:** Mohamed Baklola, Mohamed Terra, Naji Al-bawah, Husam Addin Ban Rafuid, Ziad Mohammed AL-Othrubi, Hadeel Z. Mohammed, Mohammed R. Arrabyee, Mohand Almarebi, Akram Arrabyee, Fahd Alsameeai, Sadiq Altbali, Amira Yasmine Benmelouka.

**Data curation:** Mohamed Terra, Naji Al-bawah, Mohammed R. Arrabyee, Mohand Almarebi, Akram Arrabyee, Fahd Alsameeai, Amira Yasmine Benmelouka.

**Formal analysis:** Mohamed Baklola, Mohamed Terra, Naji Al-bawah, Ziad Mohammed AL-Othrubi, Mohammed R. Arrabyee, Mohand Almarebi, Fahd Alsameeai.

**Funding acquisition:** Ziad Mohammed AL-Othrubi.

**Investigation:** Naji Al-bawah, Husam Addin Ban Rafuid, Hadeel Z. Mohammed.

**Methodology:** Naji Al-bawah, Husam Addin Ban Rafuid, Hadeel Z. Mohammed.

**Project administration:** Naji Al-bawah, Husam Addin Ban Rafuid, Hadeel Z. Mohammed, Mohammed R. Arrabyee.

**Resources:** Mohamed Baklola, Naji Al-bawah, Hadeel Z. Mohammed, Ehab Sharyan, Mohammed R. Arrabyee.

**Software:** Mohamed Baklola, Naji Al-bawah, Ehab Sharyan, Mohammed R. Arrabyee.

**Supervision:** Mohamed Baklola, Ehab Sharyan, Sadiq Altbali, Amira Yasmine Benmelouka.

**Validation:** Ehab Sharyan, Sadiq Altbali.

**Writing – original draft:** Mohammed R. Arrabyee, Mohand Almarebi, Akram Arrabyee, Fahd Alsameeai, Amira Yasmine Benmelouka.

**Writing – review & editing:** Mohamed Terra, Naji Al-bawah, Mohamed Baklola.

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
