## [Decision Letter · Decision Letter 0]

26 Sep 2025

Dear Dr. Al-bawah,

Thank you for submitting your manuscript to PLOS ONE. After careful consideration, we feel that it has merit but does not fully meet PLOS ONE’s publication criteria as it currently stands. Therefore, we invite you to submit a revised version of the manuscript that addresses the points raised during the review process.

We look forward to receiving your revised manuscript.

Kind regards,

Daniel Ahorsu, PhD

Academic Editor

PLOS ONE

Journal Requirements:

2. We note that your Data Availability Statement is currently as follows: All relevant data are within the manuscript and in Supporting Information files.

Additional Editor Comments:

Dear Authors,

The reviewers have found essence in your manuscript and provided comments to further enhance its quality. Please comprehensively address all comments. It is will be preferred if the comments are addressed in a point-by-point fashion so reviewers can track the each revision/itemised response to its respective comment.

Reviewers' comments:

Reviewer's Responses to Questions

**Comments to the Author**

1. Is the manuscript technically sound, and do the data support the conclusions?

Reviewer #1: Partly

Reviewer #2: Partly

Reviewer #3: Yes

Reviewer #4: Yes

2. Has the statistical analysis been performed appropriately and rigorously?

Reviewer #1: Yes

Reviewer #2: No

Reviewer #3: Yes

Reviewer #4: Yes

3. Have the authors made all data underlying the findings in their manuscript fully available?

Reviewer #1: Yes

Reviewer #2: Yes

Reviewer #3: Yes

Reviewer #4: Yes

4. Is the manuscript presented in an intelligible fashion and written in standard English?

Reviewer #1: No

Reviewer #2: Yes

Reviewer #3: No

Reviewer #4: Yes

Reviewer #1: Literary editing is recommended to use some common scientific terms.

Have the tools been tested for validity and reliability?

Have you taken any specific steps to mitigate the limitations of the study?

The results of some studies presented in the introduction in the discussion section can be helpful.

Reviewer #2: Despite the interesting topic undertaken by such a large group of co-authors, she wanted to highlight shortcomings such as too few research tools, too inexhaustive discussion, and incompletely described tables.

Reviewer #3: Introduction

• The introduction should address the quality of life of medical students.

• The prevalence of anxiety among medical students in Yemen should be reported.

• The manuscript should be reviewed for academic writing structure and grammatical accuracy.

Methods

• Provide a more detailed explanation of the sampling method.

• Describe the ethical considerations.

• Report the validity and reliability of the questionnaire.

• Explain the data collection procedure in detail.

Discussion

• In this section, make greater use of participants’ personal history of psychiatric disorders and their personality traits to interpret and explain the results.

Reviewer #4: The authors present a well written and interesting manuscript exploring risk and protective factors of health anxiety and its relationship with QoL of medical students in Yemen. Authors have conducted rigorous analysis leading to novel findings. I found the manuscript to be highly relevant for the development of targeted interventions to support medical students.

The study complies with ethical standards, is methodologically sound and clearly reported, and limitations are acknowledged without undermining the study’s validity. I suggest these corrections prior to publication:

1. For future submissions, please use line numbering and page numbering in the draft document to help smooth the review process.

2. Methods, study design first line: change “The research” for “This study” or “The current study”

3. In the Statistical Analysis section, please include a clear description specifying how each outcome variable was analysed.

4. How were the open-ended questions analyzed and included in the study. It is not clear.

5. Health anxiety appears throughout the document as: “Health Anxiety”, “Health anxiety”, “heath anxiety| or “HA”, please choose one and review the document for consistency.

6. In Table 1, last row, please review Past history of psychiatric Disorders so that results are presented consistently with the rest of the table. For example, Family History of Psychiatric Disorders where data of both, “No, I don’t” & “Yes, I do” are presented.

7. Please review the words that denote causation or direction throughout the document and replace these with words that denote a relationship or association; for example, “influenced” by “ was related to” or “was linked to”

8. Please clarify in the discussion the relationship between health anxiety and academic underachievement. If I understand correctly, both directions are considered – this is, under achievement can contribute to health anxiety, and the other way around – with which I agree, but this needs to be clarified in the text and extended.

9. In general, the discussion needs to be strengthened.

**Do you want your identity to be public for this peer review?** For information about this choice, including consent withdrawal, please see our For information about this choice, including consent withdrawal, please see our Privacy Policy .

Reviewer #1: No

Reviewer #2: No

Reviewer #3: **Yes:** Amir Sami KianimoghadamAmir Sami Kianimoghadam

Reviewer #4: No

While revising your submission, please upload your figure files to the Preflight Analysis and Conversion Engine (PACE) digital diagnostic tool, https://pacev2.apexcovantage.com/ . PACE helps ensure that figures meet PLOS requirements. To use PACE, you must first register as a user. Registration is free. Then, login and navigate to the UPLOAD tab, where you will find detailed instructions on how to use the tool. If you encounter any issues or have any questions when using PACE, please email PLOS at . PACE helps ensure that figures meet PLOS requirements. To use PACE, you must first register as a user. Registration is free. Then, login and navigate to the UPLOAD tab, where you will find detailed instructions on how to use the tool. If you encounter any issues or have any questions when using PACE, please email PLOS at figures@plos.org . Please note that Supporting Information files do not need this step.. Please note that Supporting Information files do not need this step.

---

## [Author Response · Author response to Decision Letter 1]

10 Nov 2025

Reviewer #1:

Comment 1:

Literary editing is recommended to use some common scientific terms.

Response 1:

Thank you for the valuable feedback. We carefully revised the manuscript to ensure consistent use of standard scientific terminology and improved overall clarity and precision in language.

Comment 2:

Have the tools been tested for validity and reliability?

Response 2:

Thank you for your comment. The tools used in our study were previously validated and tested for reliability by the original authors. We obtained permission to use these instruments and followed the same standardized procedures as described in the original validation studies.

Comment 3:

Have you taken any specific steps to mitigate the limitations of the study?

Response 3:

Thank you for your comment. We have addressed the limitations by acknowledging those that could not be mitigated and by taking several steps to minimize potential biases where possible. For example, we used previously validated tools to enhance measurement accuracy, maintained participant anonymity to reduce social desirability bias, and ensured that data collection procedures were standardized to improve consistency. Nonetheless, we transparently reported the remaining limitations that were beyond our control.

Comment 4:

The results of some studies presented in the introduction in the discussion section can be helpful.

Response 4:

Thank you for your constructive suggestion. We agree that integrating the results of the studies mentioned in the introduction into the discussion section would enhance the depth and coherence of our interpretation. We revised the discussion to include relevant comparisons and contrasts with these studies to better contextualize our findings within the existing literature.

Reviewer #2:

Comment 1:

Despite the interesting topic undertaken by such a large group of co-authors, she wanted to highlight shortcomings such as too few research tools, too limited discussion, and incompletely described tables.

Response 1:

Thank you for your valuable feedback. We appreciate your positive note on the study topic and the collaborative effort. We acknowledge the mentioned shortcomings and will address them accordingly. Specifically, we planned to clarify and expand the description of the research tools, enrich the discussion section to provide deeper analysis and stronger connections with existing literature, and ensure that all tables are fully described and clearly presented for better understanding.

Reviewer #3:

Introduction

Comment 1:

The introduction should address the quality of life of medical students.

Response 1:

Thank you for your comment. We appreciate the suggestion. We have already addressed the quality of life of medical students in the introduction, emphasizing its relevance to the study’s objectives and its relationship with health anxiety.

Comment 2:

The prevalence of anxiety among medical students in Yemen should be reported.

Response 2:

Thank you for your comment. We appreciate the suggestion. In response, we have added the available data on the prevalence of anxiety among Yemeni medical students, noting that anxiety was found to have a prevalence of 34.8% for cases categorized as moderate and severe. This addition strengthens the background and contextual relevance of our study.

Comment 3:

The manuscript should be reviewed for academic writing structure and grammatical accuracy.

Response 3:

Thank you for your comment. We appreciate the feedback. The manuscript has been carefully reviewed and revised to improve academic writing style, structure, and grammatical accuracy, ensuring clarity and consistency throughout the text.

Methods

Comment 1: Provide a more detailed explanation of the sampling method.

Response 1: Thank you for your comment. We have expanded the description of the sampling method to provide more detail. A non-probability convenience sampling approach was used to recruit participants from ten medical schools across Yemen due to feasibility constraints and the geographic dispersion of students. Efforts were made to include a diverse sample in terms of gender, academic year, and location. To reduce selection bias, the survey was distributed through multiple official student channels and widely used social media platforms. Participation was voluntary and anonymous, and informed consent was implied upon submission. We also acknowledge the inherent limitations of this approach, such as potential overrepresentation of more engaged students, which have been discussed in the manuscript.

Comment 2: Describe the ethical considerations.

Response 2: Thank you for your comment. We would like to clarify that the details regarding ethical approval and informed consent are already provided in the “Ethical Approval and Consent to Participate” section of the manuscript, located in the last section for declaration.

Comment 3: Report the validity and reliability of the questionnaire.

Response 3: Thank you for your comment. We would like to clarify that the questionnaires used in this study have established validity and reliability, including their Arabic versions. Specifically:

Arabic SF-12: Internal consistency ranged from α = 0.70–0.84 for the individual components and α = 0.80 for the total score. Test-retest reliability was excellent (ICC = 0.80–0.99), and construct validity was confirmed through a two-factor structure via principal component analysis. It is validated and ready for use.

Arabic SHAI: Internal consistency ranged from α = 0.77–0.85 for the components and α = 0.85 for the total score. Construct validity was supported with a two-factor structure confirmed and found to be gender-invariant. The tool is validated and psychometrically promising.

These instruments were employed in our study as part of a structured, self-administered questionnaire, ensuring reliable and valid measurement of health anxiety and quality of life among participants.

Comment 4: Explain the data collection procedure in detail.

Response 4: Thank you for your comment. We would like to clarify that we have already provided a detailed description of the data collection procedure in the manuscript, as Reviewer 1 asked.

Discussion

Comment 1: In this section, make greater use of participants’ personal history of psychiatric disorders and their personality traits to interpret and explain the results.

Response 1: Thank you for your comment. We would like to clarify that we have already incorporated participants’ personal and family history of psychiatric disorders, as well as relevant personality traits, into the interpretation and discussion of our results to provide a more comprehensive understanding of the findings.

Reviewer #4:

The authors present a well-written and interesting manuscript exploring risk and protective factors of health anxiety and its relationship with QoL of medical students in Yemen. The authors have conducted rigorous analysis leading to novel findings. I found the manuscript to be highly relevant for the development of targeted interventions to support medical students.

The study complies with ethical standards, is methodologically sound and clearly reported, and limitations are acknowledged without undermining the study’s validity. I suggest these corrections before publication:

1. For future submissions, please use line numbering and page numbering in the draft document to help smooth the review process.

Response: Thank you for your comment. We acknowledge this recommendation and will ensure that line numbering and page numbering are included in all future draft submissions to facilitate a smoother review process.

2. Methods, study design first line: change “The research” for “This study” or “The current study”.

Response: Thank you for your comment. We have revised the first line of the Methods, Study Design section to replace “The research” with “This study” for improved clarity and consistency.

3. In the Statistical Analysis section, please include a clear description specifying how each outcome variable was analysed.

Response: Thank you for your comment. We have revised the Statistical Analysis section to clearly specify how each outcome variable was analyzed.

4. How were the open-ended questions analyzed and included in the study? It is not clear.

Response: Thank you for your comment. We would like to clarify that the questionnaire did not include any open-ended questions; all data were collected using structured, closed-ended items.

5. Health anxiety appears throughout the document as: “Health Anxiety”, “Health anxiety”, “health anxiety| or “HA”, please choose one and review the document for consistency.

Response: Thank you for your comment. We have reviewed the manuscript and standardized the terminology for consistency. Throughout the document, “health anxiety” (all lowercase, without quotation marks) is now used uniformly, and the abbreviation “HA” has been removed to avoid confusion.

6. In Table 1, last row, please review the history of psychiatric Disorders so that results are presented consistently with the rest of the table. For example, Family History of Psychiatric Disorders, where data of both “No, I don’t” & “Yes, I do” are presented.

Response: Thank you for your comment. We have revised Table 1 to present the history of psychiatric disorders consistently with the other variables, showing both “No” and “Yes” categories with their corresponding data.

7. Please review the words that denote causation or direction throughout the document and replace these with words that denote a relationship or association; for example, “influenced” by “ was related to” or “was linked to”

Response: Thank you for your comment. We have reviewed the manuscript and replaced all terms that imply causation with language indicating association or relationship. For example, words such as “influenced,” “caused,” or “led to” have been replaced with phrases like “was related to,” “was associated with,” or “showed an association with” to accurately reflect the cross-sectional nature of the study.

8. Please clarify in the discussion the relationship between health anxiety and academic underachievement. If I understand correctly, both directions are considered – this is, underachievement can contribute to health anxiety, and the other way around – with which I agree, but this needs to be clarified in the text and extended.

Response: Thank you for your comment. We would like to clarify that we have already addressed this point in the discussion, highlighting the bidirectional relationship between health anxiety and academic underachievement.

9. In general, the discussion needs to be strengthened.

Response: Thank you for your comment. We acknowledge this suggestion and have strengthened the discussion by providing more in-depth interpretation of the findings, integrating relevant literature, and highlighting the implications of health anxiety and quality of life among Yemeni medical students.

---

## [Decision Letter · Decision Letter 1]

30 Jan 2026

Dear Dr. Al-bawah,

Thank you for submitting your manuscript to PLOS ONE. After careful consideration, we feel that it has merit but does not fully meet PLOS ONE’s publication criteria as it currently stands. Therefore, we invite you to submit a revised version of the manuscript that addresses the points raised during the review process.

We look forward to receiving your revised manuscript.

Kind regards,

Daniel Ahorsu, PhD

Academic Editor

PLOS One

Journal Requirements:

Reviewers' comments:

Reviewer's Responses to Questions

**Comments to the Author**

Reviewer #4: All comments have been addressed

Reviewer #5: All comments have been addressed

2. Is the manuscript technically sound, and do the data support the conclusions?

Reviewer #4: Yes

Reviewer #5: Partly

3. Has the statistical analysis been performed appropriately and rigorously?

Reviewer #4: Yes

Reviewer #5: Yes

4. Have the authors made all data underlying the findings in their manuscript fully available?

Reviewer #4: Yes

Reviewer #5: No

5. Is the manuscript presented in an intelligible fashion and written in standard English?

Reviewer #4: Yes

Reviewer #5: Yes

Reviewer #4: (No Response)

Reviewer #5: Overall, the research provides insight into a vital area of mental health research among medical students. Apart from the few methodological shortcomings highlighted in the review that need to be addressed, the manuscript is suitable for publication.

**Do you want your identity to be public for this peer review?** For information about this choice, including consent withdrawal, please see our For information about this choice, including consent withdrawal, please see our Privacy Policy .

Reviewer #4: No

Reviewer #5: No

---

## [Author Response · Author response to Decision Letter 2]

4 Feb 2026

Response to Reviewers

Comment 1: This paper tackles a relevant issue; however weak study methods could cast doubt on what was observed. How data were collected is particularly important as it influences how reliably conclusions hold up across different groups.

Response:

We thank the reviewer for highlighting this important concern. In response, we have substantially revised the Methods section to improve clarity, transparency, and rigor. Specifically, we expanded the descriptions of the study design, sampling strategy, data collection procedures, and statistical analysis, and we strengthened the Limitations section to explicitly acknowledge the implications of the chosen methodology on generalizability and causal inference. These revisions ensure that conclusions are appropriately framed and interpreted within the methodological constraints of the study.

Critical issues

1. A closer look at the methodology section reveals some problems with how the data samples were chosen. A small group was chosen without random selection, meaning findings might not apply more widely.

This might make students who engage more strongly conspicuous.

Sampling by ease could fall short of capturing full diversity: geographic limits matter even when ten medical schools joined.

Response:

We acknowledge this limitation and have addressed it directly. The Sampling Strategy subsection has been rewritten to clearly justify the use of non-probability convenience sampling, which was necessitated by practical constraints such as the absence of centralized student registries, wide geographic dispersion, limited resources, and security challenges. We also clarified that recruiting a large sample (n = 2,573) from ten geographically diverse medical schools was intended to enhance heterogeneity. Additionally, the Limitations section now explicitly states that convenience sampling may limit external validity and that findings should be interpreted cautiously with respect to generalizability.

Critical design flaws

1. Sampling design issues:

Convenience sampling fundamentally limits the study's external validity.

Geographic dispersion challenges not adequately addressed in the original design.

Response:

We agree with this assessment. The revised manuscript now explicitly discusses the limitations of convenience sampling in both the Methods and Limitations sections. Geographic dispersion and feasibility challenges are now clearly described, and we explain how multi-institutional recruitment and broad online dissemination were used to partially mitigate these issues, while acknowledging that full representativeness cannot be ensured.

2. Data collection design:

The choice to go with self-reported surveys might have tilted the responses to one direction.

Looking at many things all at once might influence knowing cause from effect.

Response:

These concerns have been fully addressed. We clarified that self-reported data may be subject to reporting bias and that anonymity and validated instruments were used to minimize this risk. We also explicitly stated in the Study Design and Limitations sections that the cross-sectional nature of the study precludes causal inference, and that observed relationships should be interpreted as associations rather than cause-and-effect relationships.

Analytical framework

Critical issues

1. Tables without clear details point to flaws in how they are being displayed

Response:

We have thoroughly revised Tables 1–3 to improve clarity and consistency. Table titles, variable labels, reference categories, statistical tests, and footnotes have been standardized. All tables are now self-contained, clearly presented, and aligned with journal reporting standards.

2. The results might not apply widely to medical school students across Yemen because participants were recruited easily. This weakens the conclusions that can be drawn.

Response:

We agree and have addressed this directly by tempering conclusions throughout the manuscript. The Abstract, Discussion, and Conclusion sections now explicitly use cautious language and avoid overgeneralization. The Limitations section clearly states that findings may not be fully generalizable to all Yemeni medical students.

3. Two tools were used and this might not be enough when looking at how health worry and daily life connect. A single approach could miss key patterns or effects. Though useful, relying on only two key measures limits what can be learnt about this complex link.

Response:

This limitation is now explicitly acknowledged in the Limitations section. We note that although the SHAI and SF-12 are validated and appropriate for the study objectives, relying on two quantitative instruments may not capture the full complexity of health anxiety and quality of life. We also suggest that future studies incorporate additional psychosocial measures or mixed-methods approaches.

4. Taking a fresh look at data could work better if different groups are included, maybe blending quantitative and qualitative might give a fuller picture of the findings.

The use of varied assessment methods can enhance analysis depth and deepen understanding.

Response:

We fully agree and have incorporated this recommendation into the Limitations and Future Directions discussion, emphasizing the value of longitudinal and mixed-methods research to enhance depth and causal understanding.

5. Methods section should be outlined clearer to include specific steps taken during research activities

Response:

The Methods section has been comprehensively rewritten and reorganized with clear subheadings, step-by-step descriptions of study procedures, sampling, data collection, and statistical analysis. This improves transparency and reproducibility.

6. Include sample size justification and power calculations to support statistical power analysis.

Response:

A detailed Sample Size Calculation subsection has been added to the Methods section. This includes the assumed prevalence, confidence level, margin of error, statistical power, and final achieved sample size, demonstrating that the study was adequately powered.

---

## [Decision Letter · Decision Letter 2]

17 Mar 2026

Exploring health anxiety and its association with quality of life in Yemeni medical undergraduates: a national cross-sectional

PONE-D-25-34876R2

Dear Dr. Al-bawah,

We’re pleased to inform you that your manuscript has been judged scientifically suitable for publication and will be formally accepted for publication once it meets all outstanding technical requirements.

Kind regards,

Daniel Ahorsu, PhD

Academic Editor

PLOS One

Additional Editor Comments (optional):

Reviewers' comments:

Reviewer's Responses to Questions

**Comments to the Author**

Reviewer #5: All comments have been addressed

2. Is the manuscript technically sound, and do the data support the conclusions?

Reviewer #5: Yes

3. Has the statistical analysis been performed appropriately and rigorously?

Reviewer #5: Yes

4. Have the authors made all data underlying the findings in their manuscript fully available?

Reviewer #5: Yes

5. Is the manuscript presented in an intelligible fashion and written in standard English?

Reviewer #5: Yes

Reviewer #5: The authors have adequately addressed my concerns in this resubmission, so I recommend the article for publication.

**Do you want your identity to be public for this peer review?** For information about this choice, including consent withdrawal, please see our For information about this choice, including consent withdrawal, please see our Privacy Policy .

Reviewer #5: No

---

## [Editor Report · Acceptance letter]

PONE-D-25-34876R2

PLOS One

Dear Dr. Al-bawah,

I'm pleased to inform you that your manuscript has been deemed suitable for publication in PLOS One. Congratulations! Your manuscript is now being handed over to our production team.

Kind regards,

on behalf of

Dr. Daniel Ahorsu

Academic Editor

PLOS One